# Tree Cover for the Year 2010 of the Metropolitan Region of São Paulo, Brazil

**Fabien H. Wagner \*** and **Mayumi C.M. Hirye**

Remote Sensing Division, National Institute for Space Research—INPE, São José dos Campos 12227-010, SP, Brazil; mayhirye@hotmail.com

**\*** Correspondence: wagner.h.fabien@gmail.com

**Abstract:** Mapping urban trees with images at a very high spatial resolution ($\leq 1$ m) is a particularly relevant recent challenge due to the need to assess the ecosystem services they provide. However, due to the effort needed to produce these maps from tree censuses or with remote sensing data, few cities in the world have a complete tree cover map. Here, we present the tree cover data at 1-m spatial resolution of the Metropolitan Region of São Paulo, Brazil, the fourth largest urban agglomeration in the world. This dataset, based on 71 orthorectified RGB aerial photographs taken in 2010 at 1-m spatial resolution, was produced using a deep learning method for image segmentation called U-net. The model was trained with 1286 images of size $64 \times 64$ pixels at 1-m spatial resolution, containing one or more trees or only background, and their labelled masks. The validation was based on 322 images of the same size not used in the training and their labelled masks. The map produced by the U-net algorithm showed an excellent level of accuracy, with an overall accuracy of 96.4% and an F1-score of 0.941 (precision = 0.945 and recall = 0.937). This dataset is a valuable input for the estimation of urban forest ecosystem services, and more broadly for urban studies or urban ecological modelling of the São Paulo Metropolitan Region.

**Dataset:** The dataset is available at https://doi.org/10.5281/zenodo.3373632.

**Dataset License:** CC-BY

**Keywords:** urban tree cover; urban ecosystem services; image segmentation; u-net model; deep learning

---

## 1. Introduction

Recently, urban and peri-urban forests have received growing interest because they are now officially recognized as ecosystem service providers that can help to achieve sustainable development goals (SDGs), and particularly the SDG 11, which aim to make cities and human settlements inclusive, safe, resilient, and sustainable [1,2]. These forests can provide ecosystem services related (i) to provision and wealth, as they can be used for production of food or goods; (ii) to climate-related regulation, such as the heat island effect reduction or mitigation, or runoff, which contributes to urban environment safety; (iii) to ecosystem support, such as carbon and nutrient cycling, photosynthesis and soil formation; and, finally, (iv) to cultural services, by providing an aesthetic environment for recreation spaces and social venues and improving the diversity and attractiveness of the cities by creating diverse landscapes and increasing biodiversity [3–5].

In its last report on the state of the world's forests, the Food and Agriculture Organization of the United Nations (FAO) presented a methodology to quantify the benefits of urban and peri-urban forests (see Box 21 in [1]). It consists of mapping and measuring forests and trees on the ground or in

images and using i-Tree Eco (www.itreetools.org), a software program designed to assess specific tree systems' benefits and also to express their value in monetary terms [1,6]. However, these estimates of services/disservices often use random point sampling and visual interpretation, and the outputs are not spatially explicit maps, limiting the interpretation of where ecosystem services are concentrated, and who benefits [7].

Furthermore, while such ground maps are feasible in cities or towns in the developed world, it is extremely challenging for mega-cities of developing countries, such as the Metropolitan Region of São Paulo (MRSP), which is the most important agglomeration in Brazil and the fourth largest urban agglomeration in the world with 21.6 million inhabitants [8,9]. In the MRSP, efforts have been made in recent years to map forest/vegetation patches with satellite or aerial images and to map the trees on the streets of the São Paulo municipality (both datasets available at http://geosampa.prefeitura.sp.gov.br). However, the complete tree cover map of the MRSP is still lacking.

Making tree cover maps with very high-resolution images is still challenging. It has been shown that some classical pixel or segment methods can achieve good accuracy. For example, an overall accuracy of 79.3% was achieved for the state of Wisconsin tree canopy cover with the most recent published method [7]. However, we are still far from the accuracy that can been reached by deep learning-based segmentation methods. Recently, an innovative method to extract tree cover in very high-resolution images was proposed for tropical forests [10]. Using a convolutional network called U-net [11], it was shown that forest cover could be segmented at a regional scale with very high-resolution images, and an overall accuracy >95% was obtained for the forest cover map [10], a value previously unattainable with traditional segmentation methods.

In our dataset, we provide the tree cover map for the year 2010 of the MRSP. Aerial RGB images with a spatial resolution of 1 m provided by the 'Empresa Paulista de Planejamento Metropolitano S.A' (Emplasa) were used in combination with the U-net deep learning method to segment the tree cover in the images. The dataset presented in this paper mitigates the lack of a reliable and complete tree cover map for the MRSP and can be broadly used for urban studies, urban ecological modelling and is an important contribution to the estimation of urban forests and street tree ecosystem services. The tree cover dataset at 1-m spatial resolution is available at https://doi.org/10.5281/zenodo.3373632 [12].

## 2. Materials and Methods

### 2.1. Study Site

The tree cover dataset covers the MRSP, which consists of 39 municipalities and covers a total area of ~8000 km$^2$ (Figure 1). The shapefiles containing the border of the MRSP and of the municipalities are available online (http://datageo.ambiente.sp.gov.br/app/# in the directory `/Limites Administrativos-Completo/`).

### 2.2. High-Resolution Images of São Paulo

In this study, we used 71 orthorectified RGB aerial photographs that were produced and made available to this project by the 'Empresa Paulista de Planejamento Metropolitano S.A' (Emplasa). Emplasa is a public institution that elaborates and subsidizes the implementation of public policies and integrated projects of urban and regional development in the São Paulo State. The 71 orthorectified RGB aerial photographs from 2010 are a sample of the complete set of aerial photographs of the State of São Paulo, which comprises 1727 orthophotos with 30% to 60% of lateral overlap between flights. These aerial orthophotos were generated with a spatial resolution of 1 m on average and were acquired over the region during the winter dry months (JUL-AUG-SEP) in 2010 and 2011. The aircraft used were a Carajá Turboprop and a Lear Jet, and the cameras were the Ultracam, models X and XP, acquired from Microsoft. The image can be visualized on the following Brazilian government websites: http://datageo.ambiente.sp.gov.br/app/# in the directory `/Base Imagem/Portal de imagens-DigitalGlobe/Ortofotos do Estado de São Paulo-2010/2011 (EMPLASA)` or in GIS

software using the WMS link: http://datageo.ambiente.sp.gov.br/serviceTranslator/rest/getXml/
Geoserver_Imagem/ORTOFOTOS_EMPLASA_2010/1435155780713/wms.

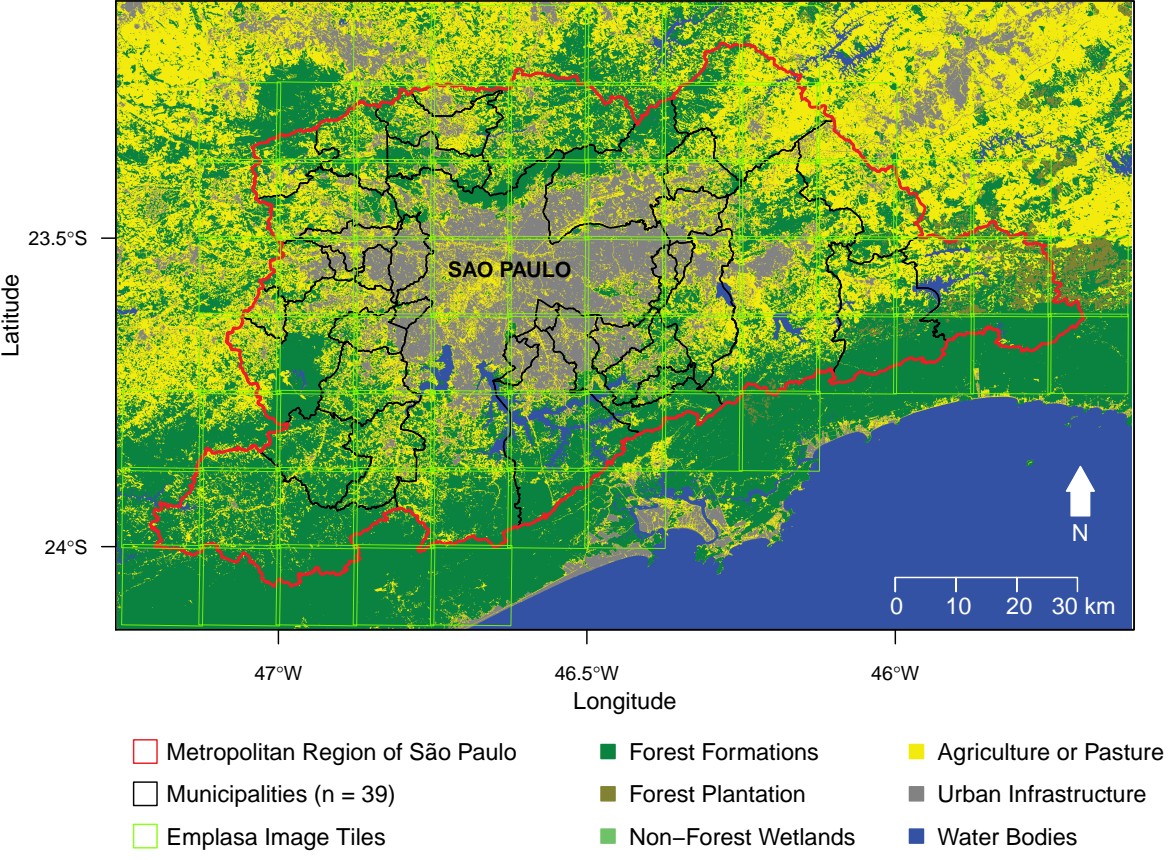

**Figure 1.** Geographical locations of the borders of the Metropolitan region of São Paulo in red;
municipality borders are represented in black; and, the extents of EMPLASA images used to generate
the tree cover mask in this study are shown in green. The background colours are the 2010 land
use/cover classes from the MapBiomas project [13]. No copyright is associated to the MapBiomas data.

*2.3. Tree Cover Segmentation*

2.3.1. U-Net Model

In this study, we used a convolutional network for image segmentation known as U-net [11].
Details regarding the architecture of the model can be found in [10]. This network performs a per-pixel
classification, predicting the probability of each pixel to belong to a particular class. This U-net model
has recently proven to become a new standard in image dense labeling [14]. We used the three-band RGB
image as the input. The code of the U-net model was adapted from the original U-net code developed
for Keras and Rstudio (available here: https://keras.rstudio.com/articles/examples/unet.html).

2.3.2. Network Training

To train the U-net algorithm to recognize and segment tree cover, a tree cover mask was manually
delineated in some parts of one of the aerial images (image ID: SF-23-Y-C-VI-2-SO), resulting in 4015
polygons. The image was chosen because it presented all different types of São Paulo tree cover,
including isolated trees, natural forests, natural degraded forests and planted forest as well as a high
diversity of background classes, with different urban building types (high-rise buildings, individual
houses, slums and industrial buildings) and other important classes for the city of São Paulo such
as water reservoirs, roads and highways. The polygons of tree cover were converted to a raster of

identical spatial resolution and dimensions as the image SF-23-Y-C-VI-2-SO, where 1 and 0 indicate tree cover and background, respectively. Then, 1608 images and their associated labels, both with a size of 64 × 64 pixels of 1-m spatial resolution, were extracted from the image SF-23-Y-C-VI-2-SO and tree cover raster. Among the extracted images, 1296 contained trees or forest and 312 contained only background. A regular grid with a cell size of 64 × 64 pixels was used to ease the extraction of these images. To constitute the training and validation image samples, among the 1608 images, we randomly selected 1286 (80%) to be used for the training and 322 (20%) for the validation. Data augmentation was applied randomly to the input images, including 0/90/180/270° rotations and changes in the brightness, saturation and hue by converting them from RGB to Brightness-Saturation-Hue space (BSH), and modulated the current values by between 95–110% for brightness, 95–105% for saturation and 99–101% for hue (as changes in the plant hues are not expected). We trained our network for 150 epochs, with 16 images per batch. In deep learning, one epoch represents a complete learning cycle, where all the training images have been presented once to the neural network. To ease computation and convergence, the training images are sent to the network in small amounts called a batch. The initial learning rate was set to $1 \times 10^{-4}$. The optimization was stopped when the loss function improvement did not exceed $1 \times 10^{-4}$.

### 2.3.3. Segmentation Accuracy Assessment

Two performance metrics were computed. First, the overall accuracy was computed as the percentage of correctly classified pixels. Second, the F1 score was computed for each class *i* as the harmonic average of the precision and recall, Equation (1), where precision was the ratio of the number of segments classified correctly as *i* and the number of all segments (true and false positive), and recall was the ratio of the number of segments classified correctly as *i* and the total number of segments belonging to class *i* (true positive and false negative). This score varies between 0 (lowest value) and 1 (best value).

$$F1_i \quad = \quad 2 \times \frac{precision_i \times recall_i}{(precision_i + recall_i)} \tag{1}$$

### 2.3.4. Prediction

For prediction, each orthophotograph was cropped on a regular grid of 512 × 512 pixels, and 64 neighbour pixels were added on each side to create an overlap between the patches. The predictions were made on the 640 × 640 pixels images, and the resulting images were cropped to 512 × 512 pixels and merged to reconstitute an image of the tree cover with the original orthophotograph extent. This overlapping method was used to avoid the artifact of prediction on the border, a known problem for the U-net algorithm [11]. The prediction returned by the algorithm is an image containing the probability of each pixels to belong to the tree cover class. The pixels with probability above or equal to 0.5 were labelled as tree cover (value = 1), and background otherwise (value = 0). The resulting tree cover map had the same 1-m spatial resolution as the input images.

### 2.3.5. Algorithm

The model was coded in R language [15] with Rstudio interface to Keras [16,17] and a Tensorflow backend [18]. The training of the models took ~10 h using GPU on a Nvidia GeForce GTX970m with 3 GB dedicated memory. Prediction of tree cover using GPU of a single image took approximately 35 min.

## 3. Results

### 3.1. Tree Cover Segmentation Details and Accuracies

The overall accuracy measured for the 322 images of the validation sample was 96.4 %, and the F1-score was 0.941 (precision = 0.945 and recall = 0.937), Table 1. Time for convergence was ~10 h.

The best model was obtained after 107 epochs with 16 images per batch (Table 1). Results of the segmentation for two image subsets of the São Paulo municipality are presented in Figure 2. Results of the tree cover area and percentage for the MRSP and all municipalities are presented in Table 2. The tree cover percentage ranges from 9.3% (São Caetano do Sul) to 97.0% (Juquitiba), and the municipality of São Paulo has a tree cover of 37.7%. Considering (i) a mean tree crown area of 32.5 m$^2$, as estimated for São Paulo previously [19,20], based on the mean tree crown area of 1109 adult trees of the species *Tipuana tipu* (Benth.) Kuntze, the most common species in the city, and (ii) the total MRSP tree cover obtained with the U-net segmentation (Table 2), we estimated that the number of trees in the MRSP is ~136,484,100.

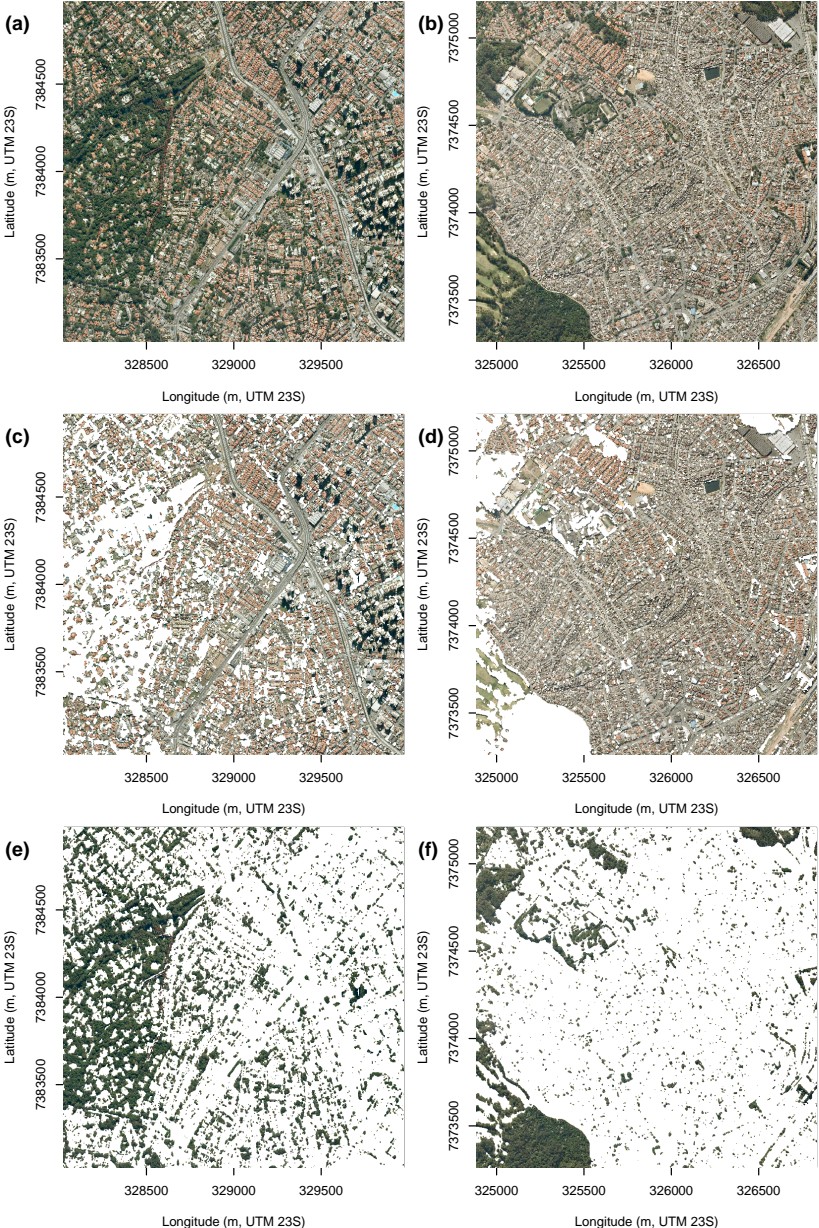

**Figure 2.** Details of the segmentation for two image subsets not used for the training. Original RGB images (**a**,**b**), original RGB image with tree mask obtained by the U-net model in white colour (**c**,**d**), only trees obtained by the U-net model segmentation and background masked in white (**e**,**f**). Each image covers approximately 4 km$^2$. The images are the property of the EMPLASA and have been made available to the authors for research purposes. No copyright is associated to these images.

**Table 1.** Numerical evaluation of the models and convergence details.

| Model | Epoch | Batch | Training Sample | Validation Sample | Overall Accuracy | F1-Score | Precision | Recall |
|---|---|---|---|---|---|---|---|---|
| Tree cover | 107 | 16 | 1286 | 322 | 96.40% | 0.941 | 0.945 | 0.937 |

**Table 2.** Area, estimations of tree cover and percent cover estimates for all municipalities of the Metropolitan region of São Paulo.

| Municipality | Area (m$^2$) | Tree cover (m$^2$) | Tree Cover Proportion (%) |
|---|---|---|---|
| ARUJA | 96,080,440 | 54,270,091 | 56.48 |
| BARUERI | 65,692,474 | 20,102,919 | 30.60 |
| BIRITIBA-MIRIM | 317,237,148 | 223,608,775 | 70.49 |
| CAIEIRAS | 96,102,694 | 71,034,715 | 73.92 |
| CAJAMAR | 131,347,100 | 86,694,991 | 66.00 |
| CARAPICUIBA | 34,548,308 | 8,292,009 | 24.00 |
| COTIA | 324,070,631 | 221,935,409 | 68.48 |
| DIADEMA | 30,791,640 | 6,030,683 | 19.59 |
| EMBU | 70,394,395 | 37,736,857 | 53.61 |
| EMBU-GUAÇU | 155,639,645 | 80,348,397 | 51.62 |
| FERRAZ DE VASCONCELOS | 29,556,363 | 12,981,528 | 43.92 |
| FRANCISCO MORATO | 49,071,419 | 25,169,178 | 51.29 |
| FRANCO DA ROCHA | 134,156,972 | 77,066,747 | 57.45 |
| GUARAREMA | 270,677,717 | 131,200,741 | 48.47 |
| GUARULHOS | 318,598,553 | 151,441,828 | 47.53 |
| ITAPECERICA DA SERRA | 150,882,196 | 97,725,433 | 64.77 |
| ITAPEVI | 82,674,965 | 43,733,761 | 52.90 |
| ITAQUAQUECETUBA | 82,577,422 | 25,236,094 | 30.56 |
| JANDIRA | 17,455,689 | 6,071,403 | 34.78 |
| JUQUITIBA | 522,311,329 | 454,485,483 | 87.01 |
| MAIRIPORA | 320,642,337 | 232,198,368 | 72.42 |
| MAUA | 61,849,444 | 20,879,373 | 33.76 |
| MOGI DAS CRUZES | 712,355,131 | 399,792,521 | 56.12 |
| OSASCO | 64,955,644 | 12,157,470 | 18.72 |
| PIRAPORA DO BOM JESUS | 108,541,021 | 69,323,286 | 63.87 |
| POA | 17,257,438 | 4,483,013 | 25.98 |
| RIBEIRAO PIRES | 99,089,353 | 64,058,945 | 64.65 |
| RIO GRANDE DA SERRA | 36,329,599 | 27,031,697 | 74.41 |
| SALESOPOLIS | 424,735,476 | 301,765,354 | 71.05 |
| SANTA ISABEL | 363,157,697 | 186,702,540 | 51.41 |
| SANTANA DE PARNAIBA | 179,960,318 | 105,001,269 | 58.35 |
| SANTO ANDRE | 175,734,910 | 94,738,489 | 53.91 |
| SAO BERNARDO DO CAMPO | 409,403,419 | 228,630,967 | 55.84 |
| SAO CAETANO DO SUL | 15,328,286 | 1,423,650 | 9.29 |
| SAO LOURENÇO DA SERRA | 186,359,362 | 155,273,390 | 83.32 |
| SAO PAULO | 1,520,949,482 | 573,864,553 | 37.73 |
| SUZANO | 206,127,277 | 99,847,434 | 48.44 |
| TABOAO DA SERRA | 20,387,896 | 4,527,716 | 22.21 |
| VARGEM GRANDE PAULISTA | 42,493,643 | 18,865,751 | 44.40 |
| TOTAL MRSP | 7,945,524,833 | 4,435,732,828 | 55.83 |

### 3.2. Limitations of the Tree Cover Dataset

The segmentation produced by the algorithm has three main known limitations. First, due to the shade of some buildings or mountains, the image may be too dark for the algorithm to recognize objects. In our dataset, the main misclassifications were associated with shade occurring in the Serra do Mar, a mountain system that follows the Atlantic coast in the south part of the image, which is outside the MRSP. Hence, we recommend using the tree cover data only inside the MRSP border. Misclassifications due to shadows cast by buildings were not frequent in our dataset but are likely to be observed in other cities, particularly in higher-density urban environments. Second, on relatively few occasions, it segments some green vegetation or algae which present a similar texture or color to that of the trees. Finally, during the conversion of the segmentation results in raster format to shapefile, the tree cover borders have been simplified to reduce the size of the data, so they have a smoother border than in the raster data. Accordingly, we recommend to use the raster rather than the shape for tree cover estimate. Both rasters and shapefiles are available in the tree cover data.

*3.3. Dataset Location and Format*

The tree cover data are available in the raster and shapefile formats at the Zenodo permanent repository of data (https://doi.org/10.5281/zenodo.3373632) [12]. The dataset is distributed in 71 tiles (EPSG:32723, WGS 84 / UTM zone 23S). The rasters contain one band with a value 1 if the pixel is tree cover and 0 otherwise. The shapefiles contain only polygons, and these polygons are the tree cover. The tree cover can represent individual trees, natural forests, natural degraded forests or forest plantations. The total size of the decompressed archives is 6.40 gigaoctets (6 Go for the shapefiles and 0.4 Go for the rasters).

**Author Contributions:** F.H.W. and M.C.M.H. conceived and designed the experiments; F.H.W. performed the experiments; F.H.W. analysed the data; M.C.M.H. contributed materials; F.H.W. wrote the manuscript, which was revised, reviewed and edited by M.C.M.H.; Funding was acquired by F.H.W., and the project was administered by F.H.W.

**Funding:** The research leading to these results received funding from the project BIO-RED 'Biomes of Brazil—Resilience, Recovery, and Diversity', which is supported by the São Paulo Research Foundation (FAPESP, 2015/50484-0) and the U.K. Natural Environment Research Council (NERC, NE/N012542/1). F.H.W. has been funded by FAPESP (grant 2016/17652-9). M.C.M.H acknowledges the support of CNPq through a doctoral fellowship.

**Acknowledgments:** We thank the Emplasa for the provision of the orthorectified aerial photographs.

**Conflicts of Interest:** The authors declare no conflict of interest. The founding sponsors had no role in the design of the study; in the collection, analyses, or interpretation of data; in the writing of the manuscript; or in the decision to publish the results. The founding sponsors had no role in the design of the study; in the collection, analyses, or interpretation of data; in the writing of the manuscript; or in the decision to publish the results.

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
