# Peer review of "Tree Cover for the Year 2010 of the Metropolitan Region of São Paulo, Brazil"

_data, 2010_

Round 1

Reviewer 1 Report

Lines 2-4. There is a scope to further improve this sentence. The reference to segmentation is out of context respect to “effort needed….” considering that maps were created with different techniques, not only segmentation.

Line 8. Not clear if 1286 are images or samples (training samples) taken by the images. Please clarify and correct.

Line 9. As above. Not clear if 322 are images or samples (test samples) taken by the images. Please clarify and correct.

Line 10. Note that is the final map or dataset that as “an excellent level of accuracy”, not the “segmentation”, this is only the technique. Please correct.

Line 11. Note that “accuracy” is repeated twice, please correct this sentence.

Lines 24-30. Please better define the context of ecosystem services (and examples reported) as classical 4 categories: provisioning services, regulating services, supporting services and cultural services (Millennium Ecosystem Assessment, 2005).

Lines 31-34. I read the cited FAO’s document but the recommendation for using i-Tree software is not present in the document. Please clarify and modify accordingly.

Line 41. “in the recent years”. Please delete “the”.

Line 46. I suggest splitting this sentence on two different one, as “For example, ….”

Line 51. As in line 10, note that is the final dataset that as an accuracy; please modify.

Line 54. MRSP. Acronym already presented, please delete “Metropolitan Region of São Paulo”.

Line 63. Acronym.

Lines 98-107. As anticipated for lines 8-9, authors should better clarify the difference among training samples, test samples and images used. Here is not clear, seem that you manage 1608 image but probably are samples taken by available images.

Line 105. “Epochs” is not clear, please better define. Not clear what the authors mean.

Lines 122.123. this sentence is not cleare, please better explain. In addition, must be inserted the minimum mapping unit of the final dataset, probably 0.5 meter considering this sentence.

Line 131 “322 images”; as above, please clarify if are samples.

Line 137. “mean crown area of 36 m2”. Not clear, please better explain crown area.

Line 138. Tipuana tipu; please insert full scientific name, as “Tipuana tipu (Benth.) Kuntze”.

Lines 137-139. This sentence should be rewritten, is not easy readable.

Table 1. The table headings must be in bold.

Lines 144-145. Please note that this is an international journal, the recommendation for readers must be rewritten considering problems for shade all over the world, not only MRSP.

Line 157. 6.40 Go? Please specify.

Reviewer 2 Report

For a ‘Data Descriptor’ manuscript, I think “Tree cover for the year 2010 of the Metropolitan Region of São Paulo, Brazil” checks all the boxes. The methodology is sound and described adequately, and the results/products are well-presented and will be useful to a broad audience, especially those interested in ecosystem services and environmental justice. There are moderate issues with grammar. I highlight some of these in my Minor Comments below, but a copyeditor will need to go through this ms carefully.

Minor Comments by line number

6 – use ‘largest’ instead of ‘biggest’; do not use the words big, bigger, or biggest

7 – 1-m spatial resolution (same with line 5); use hyphen when used as an adjective

23 – need comma after resilient; always use Oxfod comma

24 – the first two paragraphs can be merged into one paragraph

33 – need a space before benefits

37 – maps and cities should be plural

37-44 – these two paragraphs should be combined

41 – have been made

45 – maps

47 – achieved

48 – can be reached

51 – at a regional scale

88 – do you mean image density labeling?

93 – images

139 – commas are useful for long numbers to help visualize order of magnitude; same with Table 2

Round 2

Reviewer 1 Report

Corrections introduced in the text seem consistent

Author Response

Thank you very much for your review.

Best regards